# Molecular Sex Differences and Clinical Gender Efficacy in Opioid Use Disorders: From Pain Management to Addiction

**DOI:** 10.3390/ijms25179314

**Published:** 2024-08-28

**Authors:** Monica Concato, Emiliana Giacomello, Ibrahim Al-Habash, Djordje Alempijevic, Yanko Georgiev Kolev, Maria Buffon, Davide Radaelli, Stefano D’Errico

**Affiliations:** 1Department of Medical Surgical and Health Sciences, University of Trieste, 34127 Trieste, Italy; monica.concato@studenti.units.it (M.C.); egiacomello@units.it (E.G.); maria.buffon@studenti.units.it (M.B.); 2Forensic Medicine Department, Mutah University, Karak 61710, Jordan; ibrhbsh_forensic@yahoo.com; 3Institute of Forensic Medicine ‘Milovan Milovanovic’, School of Medicine, University of Belgrade, 11000 Belgrade, Serbia; djordje.alempijevic@med.bg.ac.rs; 4Department of General Medicine, Forensic Medicine and Deontology, Medical University—Pleven, 5800 Pleven, Bulgaria; drforensic@gmail.com

**Keywords:** opioid, sex differences, addiction, personalized therapeutic approach

## Abstract

Opioids have been utilized for both medical and recreational purposes since their discovery. Primarily recognized for their analgesic properties, they are also associated with the development of tolerance and dependence, contributing to a significant public health concern worldwide. Sex differences in opioid use disorder reveal that while men historically exhibit higher rates of abuse, women may develop dependence more quickly and are more susceptible to the addictive nature of opioids. This narrative review explores sex differences in opioid response in both clinical and experimental models, focusing on opioid receptor mechanisms, pain modulation, and hormonal influences. Additionally, it discusses the complexities of opioid addiction and withdrawal, highlighting sex-specific responses and the role of opioid replacement therapies. Diverse experimental outcomes, together with observational data, underscore the need for further research into sex-specific opioid biological mechanisms in a wider context, including demographic, cultural, and health-related factors. A comprehensive understanding of these complexities holds the potential to enhance personalized opioid therapies.

## 1. Introduction

Since their discovery in the 1800s, opioid derivatives have been used for both medical and recreational purposes. Opioids, well-known for their role in analgesia, are also associated with adverse events [1], including the development of tolerance and dependence, which can lead to undesired consequences of both legal and illegal significance [2].

Although opioids are considered the strongest painkillers [1], their use disorder represents a major public health concern due to the rising adverse events, along with deaths from drug overdoses [3,4]. In fact, long-term opioid administration is prone to developing adverse effects, leading to tolerance and an increased risk of dependence and overdose [1].

According to the *European Drug Report 2024*, opioids were involved in an estimated 74% of fatal overdoses, with heroin being the third most reported drug in acute drug toxicity in European hospitals in 2022. While heroin remains the most frequently used illicit opioid in Europe, its dominance is waning. Other substances, such as opioid agonists often used for treatment, or new synthetic opioids, are becoming increasingly popular. Fentanyl derivatives such as carfentanil are also a growing concern, alongside potent benzimidazole opioids such as protonitazene, metonitazene, and isotonitazene, as well as compounds containing new benzodiazepines and tranquilizers [5].

In the USA, over 564,000 opioid-related deaths were observed from 1999 to 2020 [6]. During the COVID-19 pandemic, drug overdoses reached 93,000 [7], with 107,941 drug overdose deaths occurring in 2022. Between 2021 and 2022, the rate of synthetic opioid-related deaths increased by 4.1%, while rates of heroin, natural and semisynthetic opioids, and methadone declined [8]. The Substance Abuse and Mental Health Services Administration (SAMHSA) reported that the highest rates of opioid-related emergency department visits in 2022 were among males aged 26 to 44 years, with heroin, fentanyl, and oxycodone noted as the most common substances [9].

It is worth noting that opioid use has reached significant levels as prescribed medicines for pain relief, potentially contributing to the increase in opioid dependence and related pathologies [10]. In the context of pain management, it is widely accepted that women have higher pain sensitivity and a higher risk of experiencing chronic pain conditions [11,12], which could potentially lead to higher rates of opioid addiction in women compared to men. Interestingly, although higher rates of opioid abuse have been described in men [13], research involving humans suggests that women, while developing opioid dependence at an older age [14], are more susceptible than men to the addictive nature of opioid drugs [6], experiencing an accelerated progression from initial use to dependence [14]. Not only do women tend to receive prescription pain medications more frequently, in higher doses, and for longer periods than men, but they also exhibit elevated levels of physical and mental health issues, such as major depression, related to opioid abuse in comparison to men [13]. 

Nevertheless, the recent literature on opioid prescriptions has neglected sex differences in the use of these potent painkillers [15]. Moreover, the female population has been underrepresented not only in investigations of the psychopharmacological effects of abused drugs [16] and mortality studies [3,4,6,13] but also in preclinical studies, which, although susceptible to species-specific and environmental differences, can elucidate distinct molecular mechanisms and pharmacological targets.

It is worth noting that the data mentioned above suggest seemingly contrasting behaviors: women are more prone to developing opioid dependence, while men experience more opioid-related emergencies [9]. Given this evidence, how can we reconcile both experimental and observational data present in the literature?

Beyond demographic, cultural, and health-related factors influencing different patterns of opioid use in the two sexes—such as ethnicity and family history, concurrent drug use, and coexisting medical or psychiatric conditions—this narrative review aims to deepen current knowledge on the differences in the mechanisms of opioid action between the two sexes. A better understanding of the pathways leading to opioid use disorders can contribute to targeted substance abuse management and the development of more personalized therapeutic approaches in the two sexes in the future.

## 2. Discussion

Either in the context of pain modulation or drug abuse, opioids exert their effect by targeting opioid receptors, part of the G-protein-coupled receptor family. Responding to both endogenous and exogenous stimuli, these receptors are classified into four subtypes: Mu (μ), Delta (δ), Kappa (κ), and opioid receptor-like 1 (ORL1). Each of the four subtypes possesses distinctive pharmacological importance, marked by the presence of selective agonists and antagonists. These receptors can initiate both the β-arrestin and/or the G protein pathway. While β-arrestin modulates opioid receptor signaling through desensitization and internalization, the G protein cascade can lead to different effects depending on the receptor subtype involved, regulating adenylyl cyclase, phosphatidylinositol-3 kinase, the mitogen-activated protein (MAP) kinase pathway, Ca^2+^ channels, and G-protein-coupled inwardly rectifying potassium (GIRK) channels [2,17].

Being primarily involved in antinociceptive signaling, the mu opioid receptor (MOR), delta opioid receptor (DOR), kappa opioid receptor (KOR), and opioid-like receptor (ORL) are expressed as integral membrane proteins on neuronal cells of the central nervous system implicated in descending pathways of pain modulation. Opioid receptors are also distributed in non-neuronal tissues such as neuroendocrine, immune, and ectodermal cells where they are implicated in a great variety of functions including analgesia and euphoria, as well as respiratory depression [2,17].

Traditionally, opioids and opioid derivates have been used for both pain modulation and recreational purposes. We separately discuss the evidence from the literature in the two paragraphs below.

### 2.1. Pain Modulation and Hormonal Influences

Pain stems from peripheral sensors that detect nociceptive stimuli, activating the receptors in the primary afferent fibers. These signals are then distributed to neural circuits in the spinal cord’s dorsal horn and subsequently to brain regions via the ascending pathway. In the descending pathway, nerve fibers communicate from the brain to organs via the spinal cord, resulting in a plethora of responses termed pain [18]. The endogenous descending pain modulatory circuit consists of the midbrain periaqueductal gray (PAG) and its descending projections to the rostral ventromedial medulla (RVM) and spinal dorsal horn (SDH) and is involved in the analgesic effect of opioids [19]. Interestingly, pain pathways differ between the two sexes in both their anatomic and physiologic features. PAG to RVM projections are highly represented in females but their activation relies only partially on MOR agonists [20].

These observations prompted further exploration of the detailed aspects of opioid efficacy and sex differences. Recent studies, discussed below, have delved into the effects of morphine in both acute and chronic pain, comparing various opioid agonists and their impact on antinociception and addiction in both preclinical and clinical settings.

In this context, it is worth mentioning that preclinical models should always be carefully evaluated, as data on sex differences in animal models are still under debate [21,22], and historically, the majority of preclinical studies on pain have been conducted in males since female individuals pose additional challenges in experimental design [19].

In pain conditions, it has been reported that the response to opioids, such as morphine, differs between females and males. After surgery, females require 30% more morphine than males to achieve similar levels of pain relief [23,24]. Accordingly, Sharp and collaborators [7] indicate that women are more likely to be prescribed higher doses of opioids for longer periods of time than men, leading to a greater risk of developing addiction and overdose [7].

Sex differences are attributable to variations in PAG output neurons rather than opioid-sensitive GABAergic neurons on which morphine acts. Moreover, MOR expression is higher in both the central and peripheral nervous system in male rats compared to females, leading to a greater activation of the descending pathway in males [19,24]. A smaller receptor reserve in females was also hypothesized when comparing the action of morphine and fentanyl, a high-efficacy MOR agonist drug [25].

The role of PAG in determining sex differences in nociception was also extensively investigated by Bobeck et al. [25] using both MOR agonist and non-opioid compounds. In their experiment, antinociception was greater in males following morphine, DAMGO, and bicuculline administration, but no difference was seen with the use of fentanyl and kainic acid. Direct activation of PAG produced comparable antinociception in rats of the two sexes, while inhibition of GABAergic neurons localized in the PAG induced by morphine had a greater antinociceptive effect in males [25].

Moreover, peripheral mechanism, and their interaction with opioids, can account for different opioid pain modulation in the two sexes [26]. These findings explain the greater efficacy of morphine in males in both clinical and preclinical research [19,26].

On the contrary, it should be noted that some studies have reported no differences or have produced opposite results [19]. This phenomenon could be explained by several variables, including differences between animal strains, type of agonist, modality and duration of pain, and estrous cycle [6,19,24]. In this context, morphine pain relief has also been evaluated in association with naltrexone. The effect of low doses of this opioid antagonist was examined in males and females of four different rat strains. The use of the antagonist was found to enhance opioid antinociception and to reduce the levels of morphine tolerance, with doses varying between sexes and animal strains [27].

As morphine is one of the most common medications for chronic pain, its effects were also investigated in a model of persistent inflammatory pain [24]. In animal models, the anti-hyperalgesic effects of morphine are significantly augmented by the presence of persistent inflammation in males. The process is influenced by multiple mechanisms, such as the upregulation of the excitatory glutamate receptor in the RVM, augmented MOR expression and second messenger coupling in the lumbar dorsal root ganglia, and release of endogenous opioids in PAG and RVM. In contrast, the presence of chronic inflammation does not impact the potency of morphine in females [24,28].

Sex differences in morphine analgesia were also explored in clinical settings. In contrast to previous animal studies on antinociception, they had a greater effect in women after intravenous administration, with slower onset and offset speeds, hypothesizing a pharmacodynamic origin of the differences in morphine-induced analgesia [29].

Additionally, some other drugs used in pain treatment have also been considered in experiments conducted using animal models. For example, Craft et al. [16] explored the differential efficacy of various opioid agonists, demonstrating a higher efficacy in antinociception in male rats than in females. Comparing KOR agonists and MOR agonists, they suggested that sex differences in the effects of KOR agonists could be due to spinal rather than supraspinal actions. As previously reported, they also suggested that the sex differences may be explained by pharmacodynamic variables such as different densities of opioid receptors in the brain or spinal cord [16].

Furthermore, sex differences in hyperalgesia were assessed using a capsaicin model of tonic pain [30]. Capsaicin, the pungent ingredient found in hot chili peppers, was injected into rats, producing dose-dependent thermal hyperalgesia in both sexes, with a required dose three times higher in males than in females. Low-efficacy opioids, such as buprenorphine and dezocine, were used in the same study to evaluate their antihyperalgesic effects. In general, these drugs exerted their effects equally between the two sexes, with the exception of buprenorphine, which produced a greater effect in females under certain conditions. These data contrast with previous results where thermal and mechanical essays of phasic pain showed greater effectiveness in males after mu opioids consumption [30].

In addition to sex differences in pain circuitry, sex hormones may play an important role in opioid analgesia and pain sensitivity.

We already mentioned that chronic pain disproportionately affects women, who in turn are more likely to be diagnosed and treated for chronic pain disorders [31,32]. From both preclinical and clinical studies, it appears evident that the relationship between sex hormones and opioid analgesia may be compared to a chicken–egg situation, where sex hormones participate in opioid effectiveness and in turn, pain can interfere with hormonal equilibrium.

Accordingly, research exploring hormonal influence has revealed complex interactions that vary across different experimental models and hormonal states [16,19,26].

Preclinical studies have demonstrated the influence of sex hormones on the analgesic effect of morphine on both acute and chronic pain [19,24]. According to previous observations, female and orchidectomized male rats required a tenfold higher dose of opioid agonist to achieve the same level of anti-allodynia, identifying testosterone as a crucial element in regulating MOR [26]. Co-localization of MOR and estrogen receptors in the PAG of certain experimental models is thought to be responsible for the dimorphic response to morphine. Estradiol is proposed to reduce morphine-induced hyperpolarization through the uncoupling of MOR and the G-protein-gated potassium channels, as well as decreasing the number of available opioid receptors on the cell surface, inducing MOR internalization [19].

The estrous stage did not significantly influence basal nociception or opioid antinociception in the evaluation of sex differences using opioid agonists differing in selectivity and efficacy at KOR versus MOR receptors. However, Craft et al. [16] examined randomly cycling females, sampling predominantly females in the metestrus and diestrus stages of the estrous cycle. The inclusion of different estrous stages in their study may have influenced their findings regarding sex differences in opioid effects [16].

The role of sex hormones has been evaluated in additional models. Claiborne et al. [33] focused on the sex-specific antinociceptive action of ORL activation by the endogenous ligand orphanin in the spinal cord in the rat. According to their findings, estrogen reduces antinociception in females in parallel with the physiologic hormone cycling, while testosterone is necessary for this effect in males [33]. Intact gonadal hormonal status in both sexes was also proven to be necessary in the previously mentioned capsaicin model, according to Barret et al. [30]. Gonadectomized male rats exhibited higher sensitivity to capsaicin compared to intact males, while gonadectomized females exhibited lower sensitivity compared to intact females [30].

In this context, it is worth mentioning that chronic pain has been shown to influence both the hormonal profile and the ovulatory cycle in females. In fact, in the presence of persistent algic conditions, female rats experience irregular estrous cycles, remaining in the diestrus phase. Despite low plasma estrogen levels, they might still exert sufficient influence on morphine action on MOR and subsequent signaling cascades. This stage of the estrous cycle is also associated with decreased MOR expression in several areas, including PAG [24]. As above anticipated, it can be hypothesized that pain can modulate hormonal levels that, in turn, affect opioid response.

This comprehensive exploration underscores the intricate interaction between opioids, sex differences, and hormonal factors in pain management research (Figure 1). From the distinct roles of opioid receptor subtypes to the effects of sex hormones on pain sensitivity and opioid efficacy, these findings highlight the need for tailored approaches in both experimental and clinical studies to optimize pain relief strategies. These results could be viewed from a broader perspective as they could also serve as a valuable tool in understanding compulsive drug use and pave the way for new strategies for personalized treatment of addiction.

### 2.2. Addiction and Opioid Withdrawal

Opioids are associated with a significant risk of dependence and misuse because, in addition to inducing pain relief, they can also induce euphoria and profoundly alter behaviors by impacting the reward circuitry of the brain through the inhibition of GABA neurons in the ventral tegmental area (VTA).

Neuronal mechanisms involved in opioid reward have been extensively investigated, yet the exact circuitry remains to be fully elucidated [34]. Briefly, opioid receptors are linked to drug reward and consumptive behavior [17], with the activation of mu opioid receptors (MOR) eliciting the most potent analgesic and rewarding effects. Initially, GABA–dopamine mechanisms in the VTA of the midbrain were thought to play the main role in opioid reward. However, the role of other mesolimbic circuitries and nigrostiatal dopamine circuitries has been recently investigated. Indeed, rostromedial tegmental nucleus (RMTg) → VTA → ventrostriatal and substantia nigra pars reticulata (SNr) → substantia nigra pars compacta (SNc) → dorsostriatal pathways may serve as the primary neural mechanisms responsible for opioid reward and abuse [34].

Additionally, it is known that tolerance development is caused by a functional dissociation between the receptor and some specific G proteins, leading to activation of the β-arrestin pathway, subsequent inactivation, and degradation or recycling [2,17].

As previously mentioned, deaths from opioid overdose have significantly increased in the last two years in the U.S. [8], and in 2016, the American Society of Addiction Medicine reported an increase in opiate abuse by women, with a 400% increase in overdoses compare to the 265% increase in men [35]. These data have prompted investigations into differences between sexes both in demographics and the biological mechanisms underlying opioid abuse.

In this context, a European multicentric study investigated the telescopic effect of opioid consumption. A later onset in women, together with more severe dependence, was however only observed in the first years of consumption. As drug use becomes chronic, individual variations prevent significant predictions based on sex or duration of use [36].

The difficulty in understanding the sex-dependent effects of opioids is also evidenced in the work of Gustafsson and collaborators, who compared the abuse and misuse of opioid-containing drugs in the sexes using a Netherlands database. They reported that the risk of drug dependence is higher in men, although the risk of overdose is sometimes lower [10].

Again, preclinical models can help to shed some light on this difficult subject. Actually, preclinical models have already revealed sex-specific effects in opioid seeking, craving, relapse, and withdrawal. Specifically, as stated by Cicero et al. [37], female rats tend to acquire morphine or heroin self-administration behavior more rapidly and exhibit greater motivation to self-administer opioids compared to males [37].

Sex effects on the severity of physical dependence were investigated in adult and aged male and female rats by Mohammadian et al. [38], who found some contrasting data on the differences between the two groups. The authors reported a nonsignificant difference in the severity of physical dependence on morphine between the two sexes, but they found a sex-dependent difference in the duration of morphine withdrawal and an age-dependent psychological dependence on morphine. The explanation of these findings remains unclear, but it appears that chronic administration of morphine may lead to plastic changes in the dopaminergic circuit, which has a similar biological basis in both males and females, thus neutralizing the differences between the two groups [38].

Recent studies have suggested a link between alterations in extracellular matrix (ECM) signaling in the brain and opioid use disorder, especially in cortical and striatal circuits associated with cognition, reward, and craving [4]. ECM signaling proteins such as matrix metalloproteinases and proteoglycans seem to play a direct role in opioid seeking, craving, and relapse behaviors in rodent models of opioid addiction. Furthermore, ECM, as a network of proteins, participates in both anti- and pro-inflammatory responses, such as the activation of pro-inflammatory Tumour Necrosis Factor alpha (TNF-α) signaling or the activation of Toll-like receptor 2 (TLR2) and tissue inhibitor of metalloproteinase-1 (TIMP1) in chronic opioid use followed by withdrawal periods. By playing critical roles in the pathophysiology of opioid use disorder, ECM represents a potential therapeutic target for further investigation [4]. In the future, it would be interesting to focus on extracellular matrix proteins to investigate potential sex-based differences in their expression. This could help deepen our understanding of the differences between men and women in the development of addiction.

Although morphine is the most studied substance, oxycodone also plays a significant role in the landscape of drug addiction, being significantly prescribed to high-abuse-risk patients despite the latest guidelines [39]. In research undertaken by Iyer et al. [39], where oral oxycodone consumption was examined through limited access to an oral oxycodone model, mice showed substantial dose-dependent physical dependence without a simultaneous increase in oxycodone consumption or preference, with no differences between males and females [39]. Sharp et al. [7] compared the behavior of males and females across different rat strains, confirming that sex acts as a strain-specific factor in oral oxycodone intake. Additionally, oxycodone not only affects brain function by reinforcing drug-seeking behavior but also affects pain thresholds [7]. Despite being limited in number, these studies could serve as an excellent starting point for future research, especially concerning the development of dependence, although further research is needed to better understand the phenomenon.

It is worth mentioning that opioid withdrawal phenomena in both clinical and preclinical settings may differ. In clinical settings, abstinence occurs following the cessation of drug intake, while in preclinical research contexts, withdrawal is often triggered by antagonist drugs. However, the use of opioid antagonists impacts the intensity of withdrawal symptoms and behaviors of the user, as well as the endogenous opioid system in several brain regions. Therefore, it is important to make this distinction when discussing withdrawal syndrome, as sex-linked differences could be masked by the assumption of the antagonist [3]. In opioid replacement therapy, methadone and buprenorphine are the most common treatment options. Therapeutic response, measured as time to relapse, was significantly longer in females [40].

Several works have reported differences in opioid withdrawal and replacement therapy. Santoro et al. [40] focused their research on differences in regional brain glucose metabolism in male and female rats following morphine withdrawal and subsequent methadone or buprenorphine replacement. During spontaneous withdrawal, glucose metabolism is influenced by sex and varies in different areas of the central nervous system, while methadone and buprenorphine abolish these differences in the early stages of therapy but also produce sex-specific changes [40].

Spontaneous withdrawal signs serve as an indicator of physical dependence [38]. Therefore, these parameters are investigated to better understand the differences between the two sexes. Papaleo et al. [3] investigated sex differences and the impact of morphine dosage on the manifestation and duration of specific somatic signs during spontaneous opiate withdrawal in C57BL/6J mice. The experimental model indicated sex-linked resistance to body weight loss in female animals. Additionally, data obtained for males linked body weight loss, consequent to decreased food intake, to the prior cumulative morphine exposure. Timing of expression and duration of somatic signs also showed sex-related differences, with a more severe degree of opiate withdrawal in female mice. These data contrasted with previous studies where spontaneous opiate withdrawal duration was longer in male than in female rats [3]. Other studies have also shown inconsistent results. For example, Mohammadian et al. [38] described a shorter time of morphine withdrawal in female rats. Age and strain might be the underlying factors behind these differences in this experiment [38]. Given their influence on opioid relapse, sleep disorders during opioid withdrawal in a murine model were taken into consideration by Tisdale et al. [6]. Their research proved the presence of more severe sleep disruption in females, suggesting a greater sensitivity to morphine and consequently faster escalation to dependence than males, posing the need for further investigation [6].

The opioid antagonist naloxone triggered, in both female and male mice, a strong, dose-dependent opioid withdrawal, suggesting the presence of physical dependence. DeltaFosB, a transcription factor whose expression is increased in the brain in response to drug intake and therefore implicated in drug addiction, was used to understand the neuronal activation induced by oxycodone. Mice subjected to limited access to oral oxycodone exhibited, upon naloxone challenge, an increased number of DeltaFosB-expressing cells in the nucleus accumbens core, nucleus accumbens shell, central amygdalar nucleus capsular region, and ventral tegmental area. These are considered crucial brain regions for drug abuse and reward. DeltaFosB expression is also increased in spontaneous withdrawal following chronic morphine consumption [39].

Lastly, withdrawal was also investigated in neonatal opioid exposure in mouse pups, considered an approximation of the third trimester in human gestation. Female mice are more sensitive to neurodevelopmental delays caused by neonatal opioid withdrawal syndrome (NOWS), exhibiting more pronounced neurobehavioral phenotypes compared to male mice. Borrelli et al. also investigated the effects of thermal nociception in the NOWS mouse model with precocious and increased hyperalgesia in exposed females. Different gene sets between females and males in morphine-exposed animals were also proved, suggesting an alteration in the neurodevelopmental processes and nociception after morphine consumption and spontaneous withdrawal. Finally, the study suggested the possible use of clonidine as an effective treatment for autonomic symptoms of WS by targeting norepinephrine levels [41].

This analysis delves into the multifaceted aspects of opioid withdrawal and sex differences across clinical and preclinical contexts (Figure 2). The great variability in findings across studies underscores the need for further research to elucidate these complexities. This includes exploring factors such as age, race, and neurodevelopmental impacts on withdrawal responses.

## 3. Materials and Methods

To direct future targeted therapeutic research, this review discusses the presence of sex differences in the mechanisms through which opioids act, and their implications on response and addiction. The search, conducted in PubMed in May 2024 using the terms ((sex-based) OR (sex-dependent) OR (gender-based) OR (gender-dependent)) AND ((opioid) OR (opiate)) AND (experimental models) yielded a total of 325 results. Based on the presence of details concerning biological or molecular processes related to the action of opioids, two authors independently screened and discussed the results. A total of 42 studies were selected after reading the title and abstract. After further analysis focused on the sex-dependent mechanisms of opioids, 15 studies were finally selected and are discussed below [3,6,7,13,16,24,25,26,27,29,30,38,39,40,41] (Table 1).

## 4. Conclusions and Future Directions

Based on the evidence reported, it is possible to see that sex disparities play an important role in opioid action, both in the context of pain modulation and recreational purposes. The pain model involving opioid administration has undoubtedly been the most extensively studied, providing foundational knowledge of mechanisms of action and offering a robust basis for further research.

However, a complete understanding of the biological mechanisms underlying sex differences is still not possible. Variations in experimental designs, pain models [30], dosage, administration intervals, animal species and strains, and interindividual differences can explain some of the reported inconsistent findings [21,22]. Moreover, less investigated factors such as extracellular matrix, metabolic properties, and genetic background can influence this already complex picture [4,39].

In the clinical context, while evidence shows that female individuals are more affected by chronic pain and develop an accelerated dependence onset to prescribed opioids compared to males, this does not fully explain the rapid increase in opioid dependence among women observed in the last decade, nor why males display higher rates of opioid-related emergencies during recreational use. Additional research is needed to explore the biological mechanisms of opioid use disorders. The study of mechanisms underlying human responses to opioids and sex differences should be considered within a broader context, including demographic, cultural, and health-related factors.

Nevertheless, given the steady rise in prescription opioid use, which is correlated with increased abuse and deaths [10,42], and considering that sex seems indeed to be a significant factor, special attention should be paid to treatment choice for pain modulation [43], as well as in cases of abuse and emergency situations, paving the way for increasingly personalized medicine.

## Figures and Tables

**Figure 1 ijms-25-09314-f001:**
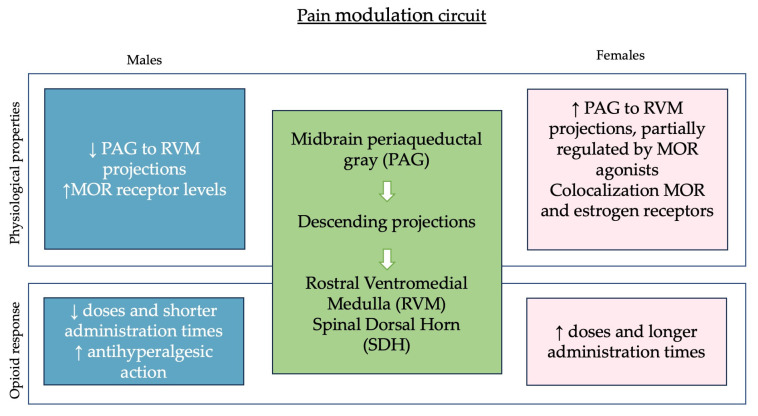
Schematic representation of sex-dependent differences in the pain modulation circuit and modulation by opioids.

**Figure 2 ijms-25-09314-f002:**
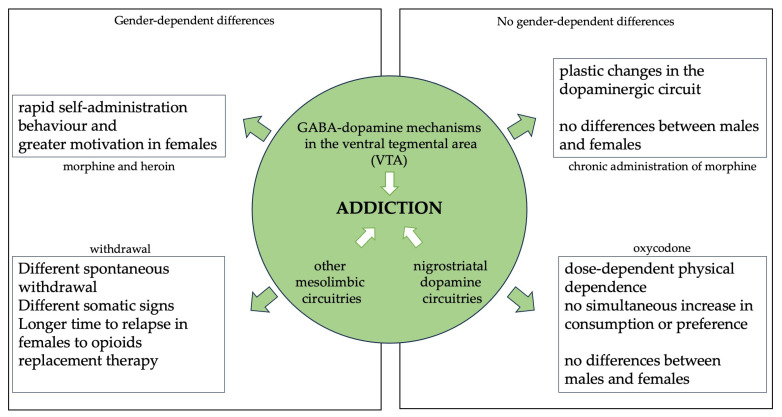
Summary of the main circuitries underlying opioid addiction and sex-related responses to selected drugs and addiction conditions arising from the literature search.

**Table 1 ijms-25-09314-t001:** Articles included in the review.

Articles	Research Subjects	Experimental Model	Results
Papaleo et al., 2006 [3]	C57BL/6J mouse	Spontaneous opiate withdrawal somatic signs —morphine	Sex- and dose-dependent (↑ female)
Tisdale at al., 2024 [6]	C57BL/6J mouse	Spontaneous opioid withdrawal —morphine	Sex-dependent effects on sleep (↑ female)
Sharp et al., 2021 [7]	Multiple rat strains	Self-administration protocol —oxycodone	Different self-administration behavior Strain-dependent
Choo et al., 2014 [13]	Human	DAWN data collection	Gender differences among opioid users seeking emergency care
Craft et al., 2001 [16]	Rat	Investigation of the antinociceptive effects of µ opioid agonists	Sex differences not related to µ opioid agonists, κ opioid antinociception may be primarily spinal rather than supraspinal
Wang et al., 2006 [24]	Rat	Thermal hyperalgesia —morphine	Sex differences in the anti-hyperalgesic action of morphine
Bobeck et al., 2009 [25]	Rat	Comparison of the antinociceptive actions of various opioids	GABAergic neuron inhibition produces greater antinociception in males
Bai et al., 2015 [26]	Rat	Orofacial persistent pain model	Sex differences in the effects of peripheral MOR agonists in long-lasting pain condition
Terner et al., 2006 [27]	Rat	Excitatory action of morphine —morphine and naltrexone	No sex or strain differences in opioid sensitivity with low doses of naltrexone
Sarton et al., 2000 [29]	Human	Pain threshold and tolerance —morphine	Sex differences in morphine potency and onset–offset speed
Barret et al., 2003 [30]	Rat	Capsaicin-induced hyperalgesia	Comparable effect of morphine
Mohammadian et al., 2019 [38]	Rat	Chronic morphine administration	Sex differences in the duration of morphine withdrawal
Iyer et al., 2022 [39]	C57BL/6J mouse	Voluntary consumption of oxycodone	No sex-dependent differences
Santoro et al., 2017 [40]	Rat	Simulation of opioid abuse and opioid replacement	Sex-dependent changes in selected areas of the CNS
Borrelli et al., 2021 [41]	Neonatal outbred mouse	Simulation of neonatal withdrawal syndrome	Sex-dependent differences, vocalization, transcriptional alterations

↑—Effects are greater in females.

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
