# Peer review of "Molecular Sex Differences and Clinical Gender Efficacy in Opioid Use Disorders: From Pain Management to Addiction"

_ijms, 2024, doi:10.3390/ijms25179314_

Round 1

Reviewer 1 Report

Comments and Suggestions for Authors

I read with interest the paper titled "Molecular gender differences in opioid use disorders: from pain management to addiction." The paper is well written and I have minor suggestions to improve the manuscript. 

1. I suggest to use the wording "sex" instead of "gender". The use of both words was made interchangeably. After reading the paper the paper is abou molecular sex differences and its not related with gender, which is a social construction. 

2. I would like to suggest two papers that could be useful in the introduction and discussion:

a. the prevalence of opioid abuse varies significantly across different regions and healthcare settings, reflecting the complexity and scope of this subject globally, as well as the substances used. Other substances as morphine and oxicodone were recently associated with misuse and abuse. Please check https://doi.org/10.3390/ph17081009

b. another publication compared sexes in the abuse and misuse from opioid-containing drugs from a database. Here, controversially, the risk of drug dependence is most of times higher in men, but the risk of overdose is sometimes lower. Please check https://link.springer.com/article/10.1007/s40264-023-01351-y

3. A lot of studies were based on animal models. I wonder to see some discussion on the similarities and differences between those models to human, and acknowledge if this is a limitation on data extrapolation. 

4. Methods. Please provide the search time limits (from-to)

5. In the conclusion, I suggest to have more focused points on what you found in your literature review. What happens between sexes in pain modulation? Whats the differences (briefly) in the mechanism between sexes? Is this contributing to abuse? Focus more on your research and less on overall thinking on the topic, which will lead to a much better conclusion. 

Author Response

I read with interest the paper titled "Molecular gender differences in opioid use disorders: from pain management to addiction." The paper is well written and I have minor suggestions to improve the manuscript. 

We thank the Reviewer for the positive comments and the points raised that allowed us to profoundly change the manuscript.

1. I suggest to use the wording "sex" instead of "gender". The use of both words was made interchangeably. After reading the paper the paper is abou molecular sex differences and its not related with gender, which is a social construction. 

The term “gender” has been changed throughout all the text.

2. I would like to suggest two papers that could be useful in the introduction and discussion:

a. the prevalence of opioid abuse varies significantly across different regions and healthcare settings, reflecting the complexity and scope of this subject globally, as well as the substances used. Other substances as morphine and oxicodone were recently associated with misuse and abuse. Please check https://doi.org/10.3390/ph17081009

We thank the Reviewer for advising article, which has been included in the Introduction (see lines 38-42)

b. another publication compared sexes in the abuse and misuse from opioid-containing drugs from a database. Here, controversially, the risk of drug dependence is most of times higher in men, but the risk of overdose is sometimes lower. Please check https://link.springer.com/article/10.1007/s40264-023-01351-y

We thank the Reviewer for advising this article, which has been included in the Introduction, in the discussion and conclusions (see lines 63, 278, 417)

3. A lot of studies were based on animal models. I wonder to see some discussion on the similarities and differences between those models to human, and acknowledge if this is a limitation on data extrapolation. 

We thank the Reviewer for rising this point. The use of animal models represents an issue in several branches of research, especially in preclinical studies aimed at being applied to humans. This point has been addressed mainly in lines 130-133. Limitations due to studies in animals have been mentioned also in other parts of the discussion and also in conclusions (see lines 403).

 4. Methods. Please provide the search time limits (from-to)

 Methods section has been modified according to Reviewer’s comments (see lines 386-393).

5. In the conclusion, I suggest to have more focused points on what you found in your literature review. What happens between sexes in pain modulation? Whats the differences (briefly) in the mechanism between sexes? Is this contributing to abuse? Focus more on your research and less on overall thinking on the topic, which will lead to a much better conclusion. 

We thank the Reviewer for raising this point. The Conclusions have been modified according to Reviewer’s comments.

Reviewer 2 Report

Comments and Suggestions for Authors

The authors performed a narrative review to explore gender differences in opioid response in both clinical and experimental models. Some major modifications are required before considering publication:

  • The current title does not fully reflect the content of the article. Indeed, the article addresses not only molecular gender differences but also clinical gender efficacy. Furthermore, it covers not only opioid use disorders but also general opioid use (pain management). Consequently, the title should be revised;
  • The authors can provide a more comprehensive introduction that contextualizes the opioid crisis and underscores the importance of gender differences;
  • The authors should expand on the methodology section to include detailed information on the specific databases used, the search terms applied, the inclusion and exclusion criteria, and the rationale behind the selection of studies. This addition will enhance the reproducibility and transparency of the review;
  • The authors should incorporate a figure to summarize the targets and mechanisms of action of opioids;
  • Additionally, the authors should include a table summarizing the included studies (detailing the year of publication, study size, main results, and other relevant information);

·         The authors should provide a more thorough discussion that critically analyzes and compares the results of different studies. They should highlight inconsistencies, explore possible reasons for divergent findings, and identify areas where evidence is lacking;

·         To finish, the authors should expand on the conclusion section to offer clear recommendations for future research and practical implications for clinical practice. They should emphasize how understanding gender differences can inform personalized opioid therapies.

Comments on the Quality of English Language

 Minor editing of English language required.

Author Response

The authors performed a narrative review to explore gender differences in opioid response in both clinical and experimental models. Some major modifications are required before considering publication:

We thank the Reviewer for the positive comments and for the points raised that allowed us to profoundly change the manuscript.

  • The current title does not fully reflect the content of the article. Indeed, the article addresses not only molecular gender differences but also clinical gender efficacy. Furthermore, it covers not only opioid use disorders but also general opioid use (pain management). Consequently, the title should be revised;

We thank the Reviewer for rising this point, the title has been modified.

  • The authors can provide a more comprehensive introduction that contextualizes the opioid crisis and underscores the importance of gender differences;

We thank the Reviewer for this comment, the Introduction has been modified according to Reviewer comments (see lines 36-44, 61-73)

  • The authors should expand on the methodology section to include detailed information on the specific databases used, the search terms applied, the inclusion and exclusion criteria, and the rationale behind the selection of studies. This addition will enhance the reproducibility and transparency of the review;

 Methods section has been modified according to Reviewer’s comments (see lines 386-393).

  • The authors should incorporate a figure to summarize the targets and mechanisms of action of opioids;

We thank the Reviewer for this comment. Opioids mechanisms of action is quite complicated, so that its graphical representation, see for example the article cited Gopalakrishnan al., 2022, in the context of this review, can result confusing. For this reason, we preferred to concisely represent opioid mechanisms of action in pain management and in drug abuse separately in figure 1 and 2.

  • Additionally, the authors should include a table summarizing the included studies (detailing the year of publication, study size, main results, and other relevant information);

 We thank the Reviewer for raising this point, we added Table 1, which summarizes the selected papers.

The authors should provide a more thorough discussion that critically analyzes and compares the results of different studies. They should highlight inconsistencies, explore possible reasons for divergent findings, and identify areas where evidence is lacking;

We thank the Reviewer for this suggestion, the Discussion has been modified according Reviewer’s comments.

  • To finish, the authors should expand on the conclusion section to offer clear recommendations for future research and practical implications for clinical practice. They should emphasize how understanding gender differences can inform personalized opioid therapies.

We thank the Reviewer for raising this point. The Conclusions have been modified according to Reviewer’s comments.

Round 2

Reviewer 2 Report

Comments and Suggestions for Authors

no further comment.

Comments on the Quality of English Language

minor editing is required